# From Conventional Therapies to Immunotherapy: Melanoma Treatment in Review

**DOI:** 10.3390/cancers12103057

**Published:** 2020-10-20

**Authors:** Lukasz Kuryk, Laura Bertinato, Monika Staniszewska, Katarzyna Pancer, Magdalena Wieczorek, Stefano Salmaso, Paolo Caliceti, Mariangela Garofalo

**Affiliations:** 1Department of Virology, National Institute of Public Health-National Institute of Hygiene, Chocimska 24, 00-791 Warsaw, Poland; kpancer@pzh.gov.pl (K.P.); mrechnio@pzh.gov.pl (M.W.); 2Clinical Science, Targovax Oy, Saukonpaadenranta 2, 00180 Helsinki, Finland; 3Department of Pharmaceutical and Pharmacological Sciences, University of Padova, Via F. Marzolo 5, 35131 Padova, Italy; laura.bertinato@studenti.unipd.it (L.B.); stefano.salmaso@unipd.it (S.S.); paolo.caliceti@unipd.it (P.C.); 4Chair of Drug and Cosmetics Biotechnology, Faculty of Chemistry, Warsaw University of Technology, Noakowskiego 3, 00-664 Warsaw, Poland; mstaniszewska@ch.pw.edu.pl; 5Centre for Advanced Materials and Technologies, Warsaw University of Technology, Poleczki 19, 02-822 Warsaw, Poland

**Keywords:** oncolytic viruses, melanoma, immunotherapy, checkpoint inhibitors, combinatory therapy

## Abstract

**Simple Summary:**

Here, we review the current state of knowledge in the field of cancer immunotherapy, focusing on the scientific rationale for the use of oncolytic viruses, checkpoint inhibitors and their combination to combat melanomas. Attention is also given to the immunological aspects of cancer therapy and the shift from conventional therapy towards immunotherapy. This review brings together information on how immunotherapy can be applied to support other cancer therapies in order to maximize the efficacy of melanoma treatment and improve clinical outcomes.

**Abstract:**

In this review, we discuss the use of oncolytic viruses and checkpoint inhibitors in cancer immunotherapy in melanoma, with a particular focus on combinatory therapies. Oncolytic viruses are promising and novel anti-cancer agents, currently under investigation in many clinical trials both as monotherapy and in combination with other therapeutics. They have shown the ability to exhibit synergistic anticancer activity with checkpoint inhibitors, chemotherapy, radiotherapy. A coupling between oncolytic viruses and checkpoint inhibitors is a well-accepted strategy for future cancer therapies. However, eradicating advanced cancers and tailoring the immune response for complete tumor clearance is an ongoing problem. Despite current advances in cancer research, monotherapy has shown limited efficacy against solid tumors. Therefore, current improvements in virus targeting, genetic modification, enhanced immunogenicity, improved oncolytic properties and combination strategies have a potential to widen the applications of immuno-oncology (IO) in cancer treatment. Here, we summarize the strategy of combinatory therapy with an oncolytic vector to combat melanoma and highlight the need to optimize current practices and improve clinical outcomes.

## 1. Introduction

Cancer is one of the three leading causes of death in industrialized countries, along with infectious and cardiovascular diseases. It is caused by the abnormal growth of the progeny of transformed cells, which have previously been subjected to mutations and several other alterations in the cell cycle and metabolism that contributed to giving these cells the typical tumor-like phenotype [1]. One of the most critical aspects in the fight against cancer is the tumor’s ability to spread in the patient’s body, even in locations far from the primary tumor location, developing metastasis [1]. This event could make the clinical picture significantly more complicated, since in order to cure cancer, all malignant cells in the patient’s body need to be destroyed and removed, preferably without side effects for the patients [2].

The immune system (IS) is a complex system which is responsible for the protection of the human body. It consists of many cell types, structures and chemical mediators with different functions that can regulate each other to work effectively and neutralize components recognized as non-self. The idea that our immune system could act as a weapon or a prevention tool against cancer cells has always been particularly attractive, especially because of the specificity of the immune response that could be elicited. The first clue about the host immune system’s alleged protective role against cancer emerged from a series of experiments on mice [3], in which it was noticed that mice previously immunized with irradiated tumor cells that were then challenged with an injection of tumor viable cells showed protection against the tumor. The same response was not observed in T cell deficient mice or mice which had been challenged with viable cells from a different tumor than the one used for the immunization process [3]. This evidence led to the discovery of the host immune system involvement in tumor-disruption and tumor prevention mechanisms, suggesting what many years of research have now shown, that is, that the host immune system has a role in the prevention and rejection of tumors [4]. However, since neither the immune system nor the tumor could be defined as simple networks, the relationship between them is obviously complex. This is due to the several factors which are involved in determining the evolution of tumorigenesis, among which there is also the immune system, which can surely exert an anti-tumor effect, but with specific subsets of immune cells, it may also perform a “foster” action on the tumor [5,6]. There are many ways that the IS could carry out its anti-tumor action. First, it protects the host from virus-induced tumors by eliminating or suppressing viral infection [7]; second, it promptly resolves inflammations, avoiding tissue exposure to an inflammatory environment, which is conducive to tumorigenesis [7,8]; third, the immune system is capable of specific recognition and disruption of tumor cells on the basis of their expression of molecules which work as antigens [7,9]. This last specific feature of IS is also known as immunosurveillance, and it is extremely important to guarantee a specific immune reaction which is directed only to tumor cells, sparing healthy tissue and avoiding many side effects [10]. This is possible because tumor cells are antigenic, meaning they express specific antigens usually called tumor associated antigens (TAAs), tumor specific antigens (TSA) or tumor rejection antigens (TRAs) [11,12,13]. The recognition and identification of these antigens is now a fundamental part in the development of effective immunotherapy, since they represent the main component with which T cells can recognize tumor cells to be activated and trigger the specific immune response. Most of the early efforts in antigens identification focused on shared tumor antigens, which could represent a valid alternative for a wide range of cancers, but these antigens are also expressed in a variety of self-tissues, leading to immunologic tolerance [14,15].

Therefore, the focus of research has slowly shifted to more tumor-specific antigens, usually generated from point mutations in normal genes, known as “neoantigens” [16,17]. Despite advances in conventional cancer therapies including chemotherapy, immuno-oncology is becoming more popular and effective in various cancer indications, including melanoma. Therefore, more conventional modalities seem to be gradually being replaced by more effective IO agents and their combinations (Table 1).

## 2. Melanoma—Epidemiology and Prevalence

Melanoma is the most aggressive type of skin cancer, and it arises from melanocytes, which are pigment-producing cells in the skin [18]. This type of cancer involves skin (mostly, but not exclusively, sun-exposed skin), but it can also occur in the eye, in the meninges and on gastrointestinal and genital mucosae [7]. In this section, we focus on cutaneous melanoma.

Melanomas can be characterized deeply from a histological point of view, thus leading to the identification of four major subtypes of melanoma [19]: Superficial spreading melanoma, nodular melanoma, lentigo malignant melanoma, and acral lentiginous melanoma. These four subtypes have different patterns of growth and come with different changes in epidermis and dermis [20]. According to a statistic evaluation carried out by the Global Cancer Observatory (GCO), which is part of the International Agency for Research on Cancer (IARC), melanoma incidence is annually increasing worldwide at a very fast rate, which in 2012 was the fastest growing of all types of cancer [21]. In GLOBOCAN 2018, the statistic evaluation of cancer incidence and mortality published by IARC, there were estimated to be approximately 290,000 new cases and 61,000 deaths related to melanoma [22], compared with the 232,000 new cases and 55,000 deaths reported in GLOBOCAN 2012. Melanoma mostly affects young and middle-aged individuals, with a median age at diagnosis of 57 years, while the incidence increases linearly from 25 years until 50 years of age, and then it decreases, especially for females [21]. Overall, the highest incidence is observed in regions with high exposure to solar radiation, such as Australia and New Zealand [23].

There are two types of risk factors for melanoma: (i) Environmental risk factors and (ii) host-related risk factors. Among the environmental risk factors commonly involved in cancer onset, for melanoma there is one particular risk factor which is deeply involved-ultraviolet (UV) light radiation from sunlight [21]. The correlation between sunlight exposure-particularly the UV-B spectrum [24]—and increased risk of melanoma has been deeply investigated, with findings that describe how exposure patterns and timing can contribute to the risk stratification for melanoma [21,25]. Intense and intermittent sun exposure is associated with a higher risk of melanoma, compared with continuous sun exposure, which is more often associated with non-melanoma skin cancers. UV-A exposure from artificial sources, such as sunbeds and devices employed in radiation phototherapy of psoriasis, is associated with a higher risk of melanoma [26]. There are a number of host risk factors related to the patient: (i) The number of congenital and acquired melanocytic nevi, which linearly correlates with melanoma incidence [21]; (ii) pigmentation characteristics of the patient, which are determined by polymorphisms in MC1R gene (melanocortin 1 receptor)—individuals with red hair, light complexion and light eyes exhibit a low pigmentation, and thus an increased risk for melanoma because of their higher sensitivity to UV exposure; (iii) family history of melanoma [21,27]; and (iv) immunosuppression, which is usually caused by comorbidities [28]. Melanoma diagnosis usually comes as an early-stage disease, in which it is possible to proceed with surgical excision and is curable in the majority of cases, while approximately 10% of patients are diagnosed at an advanced stage, which consists of an unresectable and/or metastatic melanoma [21,29]. Furthermore, stage IV melanomas are usually associated with a poor prognosis, lower probability to develop a consistent response to treatments, and, in about 30% of cases, there is brain and visceral involvement [30]. For patients with an advanced-stage melanoma, especially those who cannot undergo excisional surgery or who have metastasis, the wide range of systemic therapies represent the only way to defeat this aggressive type of cancer, which explains their importance and why they are being heavily investigated. This section provides a brief overview of the current available approaches to treat melanoma.

## 3. Conventional Cancer Therapies

### 3.1. Excisional Surgery

Surgery is taken into consideration, especially for early-stage melanomas. Excisional surgery is an effective strategy for most patients, but it is not always feasible, and in some cases (approximately 20%) the patient can present a relapse anyway, which is usually associated with a poor prognosis [31].

### 3.2. Chemotherapy

Chemotherapy for melanoma consists of the following two chemotherapeutics:Dacarbazine (DTIC): Approved by the FDA in 1975 for treatment of melanoma, it is an alkylating agent. Like every other chemotherapeutic drug, it is not highly selective for cancer cells over healthy cells, and the high number of clinical trials which have been carried out have reported a modest anti-tumor efficacy. Despite this, dacarbazine remains one of the first-line treatments for metastatic melanoma [32].Temozolomide: Despite being considered an analogue of dacarbazine, it has been studied because it has the advantage of oral administration, which is usually more versatile for the patient. Furthermore, temozolomide can reach the central nervous system and, since brain is one of the most common sites for melanoma to metastasize, this represents a crucial point for advanced melanoma treatment [32].

### 3.3. Targeted Therapies

Targeted therapies revolutionized melanoma treatment in 2011, when the first therapies were approved by FDA. They belong to the following classes:BRAF inhibitors: Since BRAF is the most frequently mutated oncogene in melanoma [33], its inhibitors have shown promising results in several clinical trials, with rapid regression of metastasis and positive responses in 50–60% melanoma patients [32,34]. The first drug belonging to this class that has been approved for melanoma is vemurafenib, a selective inhibitor of V600-mutant BRAF [33]. In a randomized phase III clinical trial (BRIM3), vemurafenib showed an objective response rate (ORR) of 48% versus 5% for dacarbazine, and a median progression-free survival (PFS) of 5.3 months versus 1.6 months for dacarbazine [33,35]. The second BRAF inhibitor came soon after the first one, with similar promising results [33]. Toxicities associated with this class of therapeutic agents include rash, arthralgia, fatigue, fever (for dabrafenib only) and photosensitivity (for vemurafenib only), but also the development of secondary non-melanoma cutaneous lesions, such as squamous-cell carcinoma [36,37].MEK inhibitors: The development of MEK inhibitors became a priority after the success with BRAF-inhibitors, and it was led by the acknowledgement that BRAF signaling is dependent on MEK1/2 downstream activation [33,38]. Trametinib belongs to this class of new targeted therapies [32], and represents the first drug of its class to be approved by the FDA as a single agent, since in the phase III METRIC clinical trial it showed an ORR of 22% and a median PFS of 4.8 months [39]. Aside from the use of MEK inhibitors to target BRAF-mutated melanomas, there is also preclinical evidence that indicates vulnerability to MEK inhibitors in a not insignificant number of melanomas which do not present BRAF V600 mutations, called wild-type BRAF melanomas (especially in NRAS-Q61-mutant tumors), and also in BRAF/NRAS wild-type melanomas, together with melanomas harboring non-V600 BRAF mutations [33,40].

A translational investigation led to evidence of a possible synergistic relationship between MEK and BRAF inhibitors. Since then, many combinatorial approaches of these two types of inhibitors have been investigated in clinical trials. The combination of vemurafenib and cobimetinib in a phase I study not only resulted in ORR and median PFS values that were very promising, but showed that the incidence of cutaneous hyperproliferative manifestations was substantially lower compared to BRAF inhibitor monotherapy [41]. The combination of BRAF and MEK inhibitors now forms the backbone of advanced BRAF-mutated melanoma treatment [33].

## 4. Cancer Immunotherapy

The goal of cancer immunotherapy is the stimulation or activation of immune responses against tumor cells, with the ultimate aim of eradicating cancer from the patient’s body (Figure 1). In the following sections, we discuss therapeutic treatments falling under the umbrella of the cancer immunotherapy field.

### 4.1. Immune Checkpoint Inhibitors (ICIs)

Immune checkpoint inhibitors are a new class of cancer therapeutics that have the physiological purpose to negatively regulate the activation of T cells. These checkpoints make it more difficult for T cells to activate, as they need both the interaction with the epitope through the MHC I class, and the presence of co-stimulatory signals to overcome the barrier of negative inhibition. Checkpoint inhibitors (CPIs) are very important to prevent continuous occurrence of immune reactions (Figure 2) [43].

The two most important immune checkpoints that have been studied in immunotherapy are the cytotoxic-T lymphocytes antigen 4 (CTLA-4) and the programmed cell death protein 1 (PD-1) [44,45]. CTLA-4 is a receptor and a member of the immunoglobulin superfamily CD28:B7 [46]. It can be found on the surface of both effector T cells and Treg cells, as its function is to regulate the extent of the early stage activation of these two types of immune cells. CTLA-4 binds CD80 and CD86 with higher affinity than CD28 does and blocks the amplification signal that the co-stimulatory binding is supposed to send, in order to trigger T cells expansion. In tumors, CTLA-4 is overexpressed to suppress the activation of immune cells which could have been successful in reaching the tumor site (generally referred to as tumor infiltrating lymphocytes—TILs) [47].

PD-1 is another co-inhibitory molecule expressed in stimulated T cells, Treg cells, B-activated cells and NK cells, and it exerts its function once it is bound to its two ligands, PD-L1 and PD-L2. PD-L1 is expressed more and is found on antigen presenting cells (APCs), dendritic cells (DCs), macrophages and B cells, but it is also expressed in tumor cells which are able to abrogate the lymphocyte response [5]. These two immune checkpoints have been investigated as a target for several monoclonal antibodies, which are already being exploited in cancer therapy for their ability in binding a specific antigen.

The first monoclonal antibody against immune checkpoints to be discovered was ipilimumab, an anti-CTLA-4 antibody that has been firstly approved as a first-line treatment of metastatic melanoma [48]. In the anti-PD-1 group there are other two common ICIs, pembrolizumab and nivolumab, both with indications for metastatic melanoma. Pembrolizumab has been the first anti-PD-1 monoclonal antibody that has been discovered, and with clinical trials KEYNOTE-001, KEYNOTE-002 and KEYNOTE-006 it has gained the first-line therapy indication for metastatic melanoma [49]. In particular, in trials KEYNOTE 006 AND KEYNOTE-002, which both presented comparative arms, patients treated with pembrolizumab significantly improved their progression-free survival (PFS), overall survival (OS) and overall response rates (ORR) relative to ipilimumab in ipilimumab-naive patients (KEYNOTE 006), and significantly improved PFS and ORR, but not OS (although OS data are immature), relative to chemotherapy in ipilimumab-refractory patients, who had also received BRAF/MEK inhibitor therapy if BRAF-mutation positive (KEYNOTE 002) [50].

Pembrolizumab can to be administered as the first line therapy (BRAF wildtype melanoma) or after treatment with ipilimumab, in a combination with anti-CTLA-4 or in patients with BRAF mutations after treatment BRAF inhibitor such as vemurafenib, sorafenib and dabrafenib. Atezolizumab in combination with cobimetinib and vemurafenib is also used for the patients with BRAF V600 mutation-positive unresectable or metastatic melanoma (first line therapy) (Figure 3) (IMspire150, NCT02908672) [51,52,53,54,55]. Therefore, it is reasonable to suppose that atezolizumab could bring some new advantages if compared to the targeting of PD-1 exerted by pembrolizumab, such as the preservation of PD-L2 interactions with PD-1 which carries out the immune checkpoint functions that avoids autoimmune reactions during therapy, allowing for a more tolerable safety profile for this immunotherapeutic new drug [56].

Initially, immunotherapy was employed in melanoma treatment with administration of interferon and interleukin cytokines, such as IFN-α and IL-2, which were approved by the FDA with melanoma indications in 1996 and 1998, respectively [32]. Unfortunately, this approach did not show notable benefits for patients, due to the severe side effects associated with systemic administration and to the much poorer therapeutic effects that came with other routes of administration, like the subcutaneous one [32,57]. A modern approach to the immunotherapy of melanoma has grown from elucidations on the role of specific immunomodulatory molecules, and led to a goal shift directed to the enhancement of cell-mediated immunity [33]. To do this, some of the aforementioned ICIs (Figure 2 and Figure 3) have been investigated and were subsequently approved for melanoma therapy:Ipilimumab (anti-CTLA-4): Gained regulatory approval by the FDA to treat melanoma after a series of phase III clinical trials (CA184-002 as a single agent, CA184-024 in combination with dacarbazine). The tumor responses according to the Response Criteria in Solid Tumors (RECIST) criteria varied from 5.7% to 11.0% in the anti-CTLA-4 treatment arms. The median overall survival (OS) was improved to 10 months for the anti-CTLA-4 monotherapy arm as compared to 6.4 months for the peptide vaccine-alone arm (HR 0.68; *p* < 0.001 [58], CA184-002, NCT00094653). The five-year survival rate was 18.2% (95% CI, 13.6% to 23.4%) for patients treated with anti-CTLA-4 + dacarbazine vs. 8.8% (95% CI, 5.7% to 12.8%) for patients treated with placebo plus dacarbazine (*p* = 0.002, CA184-024, NCT00324155) [59]. Toxicity associated with ipilimumab includes immune-related symptoms such as dermatitis, colitis, diarrhea and, less commonly, hepatitis, uveitis and hypophysitis [60].Pembrolizumab and nivolumab (anti-PD1): After the ipilimumab proof of concept that a checkpoint blockade could actually be an effective strategy to treat melanoma, pembrolizumab and nivolumab were investigated for the same indication, even if (or maybe especially because) they are selective for another receptor which is usually expressed on immune T cell surface—PD-1. Phase III clinical trial reported the median overall survival which has not been reached in the nivolumab-plus-ipilimumab group and was 37.6 months in the nivolumab group, as compared with 19.9 months in the ipilimumab group (hazard ratio for death with nivolumab plus ipilimumab vs. ipilimumab, 0.55 [*p* < 0.001]; hazard ratio for death with nivolumab vs. ipilimumab, 0.65 [*p* < 0.001]). The overall survival rate at 3 years was 58% in the nivolumab-plus-ipilimumab group and 52% in the nivolumab group, as compared with 34% in the ipilimumab group (NCT01844505) [33,61,62,63].

### 4.2. Oncolytic Virotherapy

Oncolytic virotherapy [64,65,66,67,68,69,70] has elicited increased interest over recent years, even though the first encouraging evidence that led to its development date back to the beginning of the 20th century. It consists of the employment of naturally occurring viruses (e.g., enteroviruses, reoviruses, vaccinia virus) [71,72,73,74] or genetically modified viruses (e.g., HSV, adenoviruses). Oncolytic viruses (OVs) [75,76,77] have the fundamental feature of tumor specificity and many other important advantages, such as the ability to trigger anti-tumor immune responses and the possibility to deliver specific genes in the tumor microenvironment [78].

To avoid damage in healthy tissues, oncolytic viruses are usually genetically modified so that they can replicate only in tumor cells. Their design benefits from a deletion of 24 base pairs in the viral E1A gene which makes the expressed mutated E1A protein unable to bind to retinoblastoma protein (pRb). This interaction is needed in normal cells to activate the E2F transcription factor, which leads to induction of the S-phase of the cell cycle. The deletion-bearing virus is able to infect normal cells but its replication is restricted due to the dysfunctional E1A [74]. The viruses bearing the 24 bp deletion in their E1A gene are commonly tumor-selective and referred to as ∆24-viruses. The only cells in which ∆24-viruses can replicate are tumor cells, which are usually deficient of pRb. Taken together, it is worth highlighting that oncolytic viruses work as anti-tumor agents in a two-step manner: The first is the lysis of tumor cell they have previously infected, but not before they have finished their replication cycle, so that with cell death the release of new progeny occurs. Another feature of OVs is the ability to selectively replicate in cancer cells [79,80,81]. Even though the virus can enter both healthy and cancerous cells (the selective cell entry must not be confused with exclusive cancer cell entry), there are inherent abnormalities in cancer cell pathways concerning homeostasis, response to stress and their anti-viral machinery, which can give OVs a selective advantage for their replication in these cells [82].

The anti-viral machinery in normal cells is activated by a series of pathways:Toll-like receptors (TLRs): This pathway is activated by pathogen-associated molecular patterns (PAMPs), which consist of elements of viral capsid, DNA, RNA and viral proteins. These elements are recognized by TLRs, and they stimulate the innate immune system through a variety of signaling factors (MYD88, TRIF, IRF7, IRF3, NF-kβ), leading to the release of pro-inflammatory cytokines and local type I interferon (IFN-I) [82,83].RIG-1 pathway: This pathway is activated by the detection of viral dsDNA and uses some of the same factors exploited by the TLRs pathway, such as IRF3/7. It leads to the release of IFN-I [6].IFN-I pathway: This is activated by the local production of type I interferon. After IFN-I binds to its receptor, IFNR, a cascade of signals is triggered and, through the JAK-STAT pathway, it leads to the upregulation of cell-cycle regulators such as PKR and IRF7. These two factors are important in order to contain viral spread because they induce abortive apoptosis, which blocks the replicative cycle of viruses before the viral progeny is ready to be released [82].

Conventional cytotoxic therapies, as we have already pointed out, are not always effective in melanoma patients, and this statement comes with an even heavier burden when it comes to patients with advanced (unresectable and/or metastatic) melanoma, for whom excisional surgery is not an option [21]. Within this framework, oncolytic viruses pose as a potentially valid therapeutic option for these patients, thanks to their ability to selectively target cancer cells and simultaneously trigger the patient immune system against melanoma cells [84,85]. A decisive role in the efficacy of oncolytic viruses against tumors is covered by their stimulation of immune system, which is triggered to develop a specific anti-tumor immune response by the OV. It is for this reason that the immunogenicity of melanoma as a tumor is an important feature to describe.

Melanoma has been considered an immunogenic malignancy for a long time [86,87]. Virtually all of the major enlightenments concerning tumor immunology have been experimentally observed in melanoma models. When we say that melanoma is a strongly immunogenic malignancy, we refer to the fact that it has a close relationship with host immune cells, which usually infiltrates the tumor microenvironment [87]. The distribution, density, profile and activation state of immune cells which are part of TILs can be variable and modulates the clinical outcome in melanoma patients. TILs are now recognized as an independent prognostic biomarker for melanoma, and the assessment of its composition is even more appealing because it could provide new molecular targets and biomarkers to predict therapeutic responses of immunotherapy drugs [88]. The major components of TIL infiltrate are CD8+ T lymphocytes, Tregs, NK cells, dendritic cells, and macrophages. Furthermore, the high immunogenicity of melanoma also implies the presence of a plethora of tumor antigens, which can be classified as TAAs, which are antigens located on tumor cells’ surfaces, and TSAs or neoantigens, which are more specific for a single tumor [87]. Talimogene Laherparepvec (T-VEC), also known as Imylgic or OncoVex^GM-CSF^, is the first oncolytic virus that has been approved by the FDA and the EMA to treat cancer (Figure 2). The FDA approved T-VEC in 2015, with an indication for local treatment of unresectable, subcutaneous, cutaneous and nodal lesions in patients with melanoma recurrent after initial surgery [89,90]. From this perspective, T-VEC represents a valid second-line treatment for patients with metastatic, unresectable melanoma, especially for those with stage IIIB, IIIC and IV melanoma [89].

## 5. Combinatorial Approaches with OVs in Melanoma Treatments

### 5.1. OVs with Immune Checkpoint Inhibitors

The idea behind this combinatorial approach is that these two therapeutic tools can improve each other by addressing one another’s shortcomings. Oncolytic viruses present some limitations related mainly to antiviral immunity, which makes it challenging to exploit the bloodstream to reach distant metastatic sites [91]. Therefore, triggering a tumor-specific adaptive immune response is a fundamental feature as OVs cannot travel inside the body to reach other sites, while T cells that have been sensitized to tumor cells surely can, thus assuring an antitumor response even in different sites to that of the primary tumor [91]. From this perspective, ICIs help guarantee the correct activation of the immune system, targeting specific molecules expressed either on the tumor or on immune cells (CTLA-4, PD-1, PD-L1), while viral infections obtained using OVs makes the TME more immunogenic, creating a microenvironment in which ICIs are known to work much better [92].

This combinatorial approach has been explored in numerous clinical trials [91], among which was a phase II clinical trial with 198 stage IIIB–IV melanoma patients, which was organized to evaluate: (1) ipilimumab as a monotherapy and (2) ipilimumab combined with T-VEC [92]. The results showed that the objective response rate of the combination therapy was 39%, while ipilimumab alone had an objective response rate of 18% [91,92]. T-VEC has also been investigated in combination with other ICIs such as pembrolizumab, an anti-PD-1 antibody. In the phase IB portion of the clinical trial Masterkey-265, T-VEC was administered to 21 patients with stage IIB and IV melanoma in combination with intravenous pembrolizumab. Among the criteria that have been evaluated, the safety profile of the combination was favorable, with no dose-limiting toxicities, and the objective response rate was 62%, while 33% of patients showed a complete response [92].

Multiple adenoviruses are undergoing clinical and preclinical testing in combination with ICIs, both in melanoma and other types of tumor.

The Hemminki group exploited a murine model of melanoma to establish the mechanism under the combination of the anti-PD-1 antibody with the oncolytic viruses encoding for TNFα and IL-2 [93]. What emerged from the combination therapy was a marked increase in intratumoral CD8+ T cells and a statistically significant tumor growth suppression, along with increased survival in animals. Researchers reported complete tumor regression after the course of the combinatory therapy. This preclinical research provides the rationale for a clinical trial where oncolytic adenovirus coding for TNFa and IL-2 (TILT-123) is used in melanoma patients receiving an anti-PD-1 antibody NCT04217473) [92,93].

Thomas et al. reported development of a new fusion-enhanced oncolytic immunotherapy platform based on herpes simplex virus type 1. Researchers developed various oncolytic vectors expressing e.g., GMCSF, an anti-CTLA-4 antibody-like molecule. Anti-cancer assessment was performed in vivo and in nude mouse xenograft models (melanoma, lymphoma, gliosarcoma). The combination therapy with the virus expressing GALV-GP-R- and mGM-CSF and an anti-murine PD1 antibody showed improved anti-tumor effects compared to the control. The treatment of mice with derivatives of this virus coding for anti-mCTLA-4, mCD40L, m4-1BBL, or mOX40L showed enhanced anti-cancer efficacy in un-injected tumors (abscopal effect) [94].

Also, in our previous study we have investigated the anti-cancer potency of ONCOS-102 and pembrolizumab in the humanized melanoma mouse model. Humanized mice engrafted with A2058 melanoma cells showed significant tumor volume reduction after ONCOS-102 treatment. The combination of anti-PD1 with the virus further reduced tumor volume, while pembrolizumab alone did not show therapeutic benefit by itself [45]. Systemic abscopal was also observed when combining oncolytic adenovirus and checkpoint inhibitor in a humanized NOG mouse model of melanoma [44]. These data support the scientific rationale for the ongoing clinical study of combination therapy of ONCOS-102 and pembrolizumab for the treatment of melanoma (NCT03003676).

Currently, there are many oncolytic vectors are under development and investigation in melanoma: coxsackieviruses, HF-10, adenoviruses, reoviruses, echoviruses, and Newcastle disease viruses. Therefore, it is probable that oncolytic vectors will have long-term application in the treatment of advanced melanoma not only as a monotherapy but as a part of combinatory therapies. [95].

T-VEC is the first oncolytic vector approved for the melanoma treatment. Reported data have shown improved therapeutic responses to T-VEC in combination with immune checkpoint blockade in patients with melanoma without additive toxicity [96]. T-VEC combined with anti-PD-1 based immunotherapy for unresectable stage III-IV melanoma showed an overall response rate for on-target lesions of 90%, with 6 patients resulting in a complete response in injected lesions (NCT02263508) [97]. Also, the treatment with T-VEC in patients with advanced melanoma with disease progression following multiple previous systemic therapies (vemurafenib, metformin, ipilimumab, dabrafenib, trametinib, and pembrolizumab) showed signs of anti-cancer effect, and provides potential clinical and immunotherapeutic utility of T-VEC application [98].

CAVATAK, an oncolytic immunotherapy, is an oncolytic strain of Coxsackievirus A21 (CVA21). The virus infects ICAM-1 expressing tumor cells, resulting in cell lysis, and anti-tumor immune response. The Phase II CALM study investigated the efficacy and safety of CVA21 in patients with advanced melanoma (NCT01227551). The treatment with CAVATAK resulted in elevation of the immune CD8+ T cell infiltrates within the tumor (5 of 6 patients), and increased expression of PD-L1+ cells. It was also reported that the virus was able to reconstitute immune cell infiltrates in lesions resistant to immune-checkpoint blockade [99]. The combinatory therapy trials have been conducted where CAVATAK was administered with ipilimumab (NCT02307149) or pembrolizumab (NCT02565992). The treatment with CAVATAK and anti-CTLA-4 has shown durable response with minimal toxicity. The preliminary ORR rate for the ITT population of 50.0% is higher than published rates for either agent used alone (CAVATAK: ~28% and ipilimumab: ~15–20%) in advanced melanoma patients [100]. Among the evaluable patients (intratumoral CAVATAK and systemic pembrolizumab in advanced melanoma patients), the ORR was 73% (8/11). The DCR (CR + PR + SD) was 91% (10/11). In patients with stage IVM1c disease, the ORR and the DCR is 100% (5/5). Combination therapy of the virus1 and anti-PD1 may present a new strategy for the treatment of patients with injectable advanced melanoma (CAPRA clinical trial) [101].

Another oncolytic adenovirus that has been investigated in combination with pembrolizumab is ONCOS-102 (AdV5/3-Δ24-GM-CSF), which is now under clinical trial (NCT03003676) to investigate its safety and efficacy, supported by preclinical data showing increased CD8+ T cell infiltration in tumor mass upon viral administration [92]. The therapeutics efficacy and safety of the virus was previously tested in C1 study (NCT01598129). The treatment with the virus was safe and well tolerated at the tested doses. Therapy resulted in infiltration of CD8+ T cells to tumors and up-regulation of PD-L1, highlighting the potential of ONCOS-102 as an immunosensitizing agent for combinatory therapies with checkpoint inhibitors [102]. Therefore, providing a scientific rationale for the combinatory therapy with CPIs.

To date, approximately one third of all clinical trials concerning OVs have investigated a combinatorial approach with at least one ICI [91]. Therefore, it is expected that oncolytic viruses have the capability to promote a ‘hotter’ immune microenvironment which can improve the efficacy of ICI [103,104]. Oncolytic viruses can be thought of as matches—they can light up a fire inside the tumor and this fire will make the TME “hot” enough for ICIs to strike a blow. Many clinical and preclinical models of melanoma and other solid tumors have provided strong evidence that the infection of tumor cells with OVs can result in the creation of a pro-inflammatory tumor microenvironment, which in turn translates into a new influx of T cells that can be protected from inactivation by ICIs [104,105]. Furthermore, some adenoviruses administered in combination with ICIs have been reported to boost release the pool of tumor neoantigens which can be recognized by CD8+ T cells [106]; this is a particularly important finding, because OVs (both as monotherapy and in combination) have most difficulty affecting low mutational burden cancers, which typically have a very small number of TAAs [91].

Nowadays, one of the major challenges for researchers investigating this field is to assess not only which combinations are most effective, but the dosing regimens and schedules to adopt to maximize the synergy and minimize the side effects [91,92]. This is why further clinical trials results are so impatiently awaited. ICIs have contributed to revolutionize cancer treatment. Nevertheless, the best response rates to these agents do not exceed 35% to 40% [107]. Therefore, the goal of combining OVs with ICIs is to enhance clinical efficacy. Oncolytic vectors are used in order attract the immune cells into the lesion, prime anti-tumor immune responses by development of innate and adoptive anticancer immunity. In turn, CPI therapy will prevent inhibition of activates cancer specific T cells. It is expected that those two agents can result in synergistic or additive anti-cancer effect. Interestingly, it has been demonstrated that local OV injection can modulate tumor-specific CD8^+^ T-cell responses rendering distant tumors susceptible to immune checkpoint inhibitor therapy [108]. Therefore, due to the preclinical success of this combination therapy, there is huge interests in clinical trials: results obtained from patients who have progressed after immune checkpoint inhibition (e.g., NCT 03003676) could shed the light on OV’s role in overcoming resistance to immunotherapy. By elucidating the potential of the combination of OVs and checkpoint inhibitors, further development in treatment regimens employing these novel therapeutic agents could be beneficial for patients.

Apart from combinatorial strategies, another aspect concerning the use of ICIs is often investigated to reach some improvement—the response predictions with biomarkers. There are several biomarkers associated with the response of ICIs, some of which have been approved and are currently being exploited to predict the response rate in patients before treatment begins, while others are under further study to establish whether they have a strong correlation with the extent of patients’ responses to ICIs. The most important predictive biomarker for anti-PD-1/PD-L1 antibodies is PD-L1 expression [41,82], which is evaluated by immunohistochemistry. PD-L1 expression by cancer cells is recognized as both a prognostic and predictive biomarker in patients with cutaneous melanoma. Approx. 35% of cutaneous melanomas express PD-L1, The PD-L1 immunohistochemistry (IHC) has been approved by FDA as a complement diagnostic to select patients with non-small-cell lung carcinoma (NSCLC) suitable for pembrolizumab therapy. Nevertheless, absence of PD-L1 does not necessarily translates into a poor response to anti-PD-1/PD-L1 inhibitors. Some patients with low PD-L1 expression exhibits clinical efficacy. However, further efforts are still needed to improve the clinical use of PD-L1 expression as biomarkers Apart from combinatorial strategies, another topic concerning the use of ICIs is often investigated to study prediction biomarkers. There are several biomarkers associated with the response of ICIs, some of which have been approved and are currently being exploited to predict the response rate in patients before treatment begins, while others are under further study to establish whether they have a strong correlation with the extent of patients’ responses to ICIs. The most important predictive biomarker for anti-PD-1/PD-L1 antibodies is PD-L1 expression [41,109], which is evaluated by immunohistochemistry and is a prerequisite for treatment with drugs such as atezolizumab or pembrolizumab. However, this biomarker may not be enough to identify all of the patients who could benefit from this type of therapy, and this observation led scientists to begin further studies to find more appropriate predictive biomarkers. This biomarker’s use is already well established in conventional chemotherapy regimens, but recent studies suggest that it could be exploited to predict the response to immunotherapy and, most importantly, that it could also help discriminate real disease progression from pseudo-progression in patients treated with immunotherapy, avoiding re-biopsy in these patients [109,110,111].

### 5.2. OVs with Chemotherapeutic Agents—Future Prospects

In the last decade, many preclinical models have demonstrated that chemotherapeutic agents and OVs could work synergistically [11]. There are two main approaches to setting this combination:Use OVs as adjuvant to chemotherapy, which is the most clinically relevant approach, since chemotherapeutic agents represent the cornerstone of almost every cancer standard-of-care therapy [78].Use chemotherapeutic drugs to counteract or inhibit factors that limit the effectiveness of oncolytic virotherapy such as large tumor size, poor vasculature, elevated interstitial pressure and other physical barriers [112].

It is important to consider that even if chemotherapy and OVs could seem good partners, not all combinations have showed synergistic effects. In fact, the result depends on different factors including OVs strain, cancer type and the exact drug(s) used, as well as their dosing regimen and schedule [113].

In terms of the two standard-of-care chemotherapeutic agents for melanoma, temozolomide and dacarbazine, there are both in vitro and in vivo studies that explored combinations with various oncolytic vectors. Specifically, an in vitro/in vivo study that tested the combination of dacarbazine with ZD-55-IL18, an oncolytic adenovirus encoding for IL-18, showed that there is a synergy between these two agents, observed in the induction of apoptosis of tumor cells, and inhibition of angiogenesis and metastasis [113]. Another in vitro study conducted on melanoma cell lines treated with temozolomide (TMZ) and another oncolytic adenovirus (Ad5/3.2xTyr) reported that TMZ enhanced the OV’s antitumor effect without altering the expression of CAR or other viral receptors on cancer cells, but rather by blocking the tumor cell’s cycle in the S/G2 phase, providing a better intracellular environment for the viral replicative cycle to develop. This finding is consistent with the higher number of genome copies detected inside infected tumor cells [114].

By way of conclusion, we could say that the combinatorial approach based on chemotherapy and oncolytic viruses is promising, but like every other approach, it has to face some challenges to push researchers even further. It is clear that there are some incompatibilities between chemotherapy and OVs [114] which must be taken into account when designing new combinations:Many chemotherapeutic agents induce apoptosis in cancer cells, while OVs need actively dividing cells to complete their replicative cycle successfully;Other chemotherapeutic drugs target angiogenetic mechanisms to impair tumor expansion, but this would also affect viral trafficking inside the tumor mass;The immune modulation exerted by some chemotherapeutic drugs could dampen the antitumor immune response triggered by OVs.

We must consider all of these notions to reach a combination that can work effectively with a synergistic interaction.

### 5.3. OVs with Radiotherapy—Future Prospects

Anticancer synergistic interaction of radiation and OVs therapy has solid backing in the literature, and the enhancement of viral replication due to radiotherapy has been reported in different in vitro and in vivo models such as lung cancer and pleural mesothelioma [115,116]. In melanoma, apart from the clear and demonstrated efficacy of both approaches as a monotherapy to kill cancer cells [32], the synergy is due to three other aspects:Radiotherapy may reduce the internal pressure within the tumor mass, making it easier for the OV to penetrate it and work properly.Some OVs, such as vesicular stomatitis virus (VSV) or HSV, are able to preferentially target Ras-mutated cancer cells (Ras is one of the driver mutations in melanoma). Since Ras mutations in cancer cells are associated with resistance to radiotherapy, OVs which can target these cells will exert a complementary therapeutic effect to radiotherapy.Infection of melanoma cells by OVs will lead to a release of cytokines like TNFα or TRAIL, which can sensitize tumor cells to radiation therapy.

Twigger et al. tried to combine an oncolytic reovirus with radiation therapy in a variety of melanoma cell lines, observing that the combination yielded a statistically significant enhancement of viral cytotoxicity without affecting reoviral replication rates, but with an increase in apoptosis of cancer cells [117]. In another preclinical study, Kyula et al. investigated the combination of an oncolytic *Vaccinia* virus and radiotherapy in BRAF-mutated, Ras-mutated and wild type melanoma cell lines. Results showed that in melanoma cells that carried V600D or V600E BRAF mutations there had been an increased apoptosis [42]. Also, the combination of reovirus and radiation has shown to increase the tumor growth delay of the melanoma xenografts in the treated animals, and significantly improve the overall survival rate compared to the treatment with either of the individual therapies [118]. Importantly, Ras mutation is one of the driver mutations for melanoma and is associated with radio-resistance [58]. However, some viruses like: reovirus, VSV and HSV have been able to selectively target the Ras mutated melanoma cells and mediate cell death [119]. Therefore, oncolytic vectors able to lyse the radiation-resistant melanoma cells can exhibit a complementary therapeutic effect to radiotherapy. There are many ongoing attempts to find the optimal way to combine these two strategies to maximize the antitumor effect preclinically. More investigations are needed to understand how to exploit this combination in the complex context of metastatic unresectable melanomas and their application in clinics.

## 6. Conclusions

The discovery of T cell checkpoint inhibitors and oncolytic virotherapy has changed the paradigm of oncologic treatment for some cancer types and showed a transition pattern from conventional therapies towards immuno-oncology. Oncolytic viruses can induce anti-tumor immunity and lead to the infiltration of TILs. In turn, a checkpoint blockade can prevent inhibition of T cell activity. Therefore, the combination of those agents seems to be a potent treatment regimen to combat immunogenic cancer types such as melanoma.

## Figures and Tables

**Figure 1 cancers-12-03057-f001:**
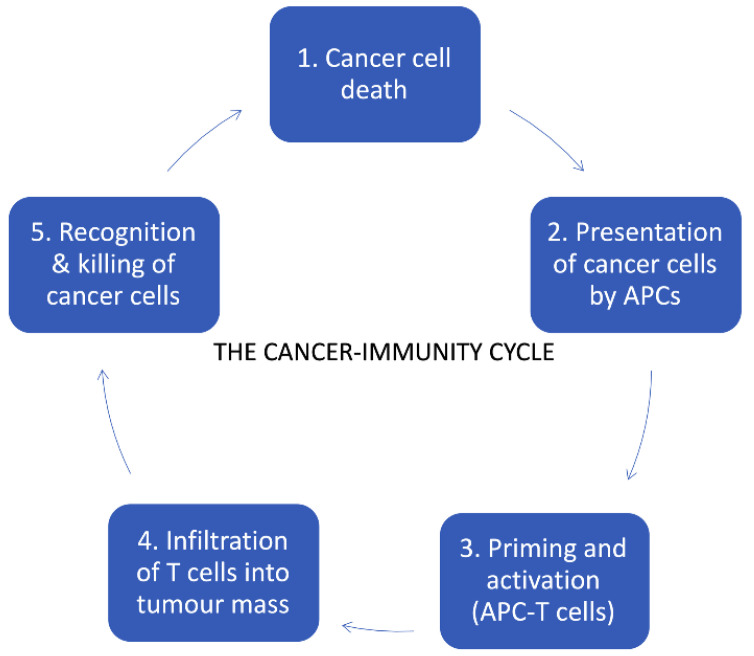
The cancer immunity cycle, modified from [42].

**Figure 2 cancers-12-03057-f002:**
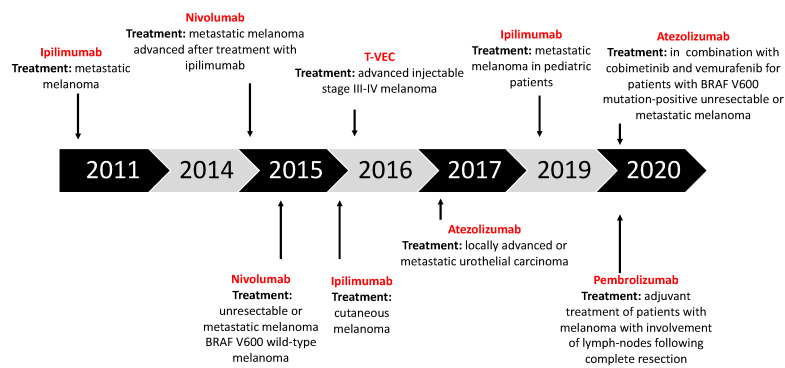
Timeline of immuno-oncology (IO) agents approved for cancer therapies.

**Figure 3 cancers-12-03057-f003:**
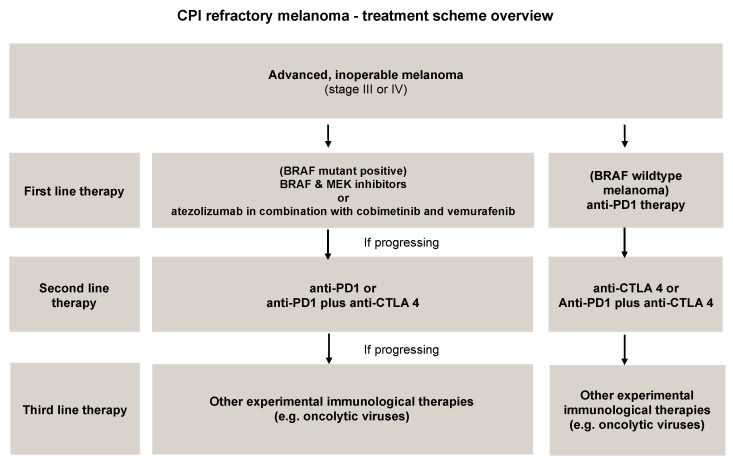
CPI refractory melanoma-treatment scheme overview.

**Table 1 cancers-12-03057-t001:** The combinatory therapy of oncolytic vectors and CPIs for melanoma treatment.

OV	Checkpoint Inhibitor	Indication	Response Data	ClinicalTrials.gov ID
**T-VEC**	Ipilimumab	Melanoma	ORR 39% (comb.) vs. 18% (ipi alone)	NCT01740297
**T-VEC**	Pembrolizumab	Stage IIIB–IV melanoma	48% ORR	NCT02263508
**T-VEC**	Pembrolizumab	Stage III–IV melanoma	N/A	NCT02965716
**HF-10**	Ipilimumab	Melanoma	N/A	NCT031530085
**HF-10**	Ipilimumab	Melanoma	BORR 24% (at 24 weeks); median PFS 19 months; median OS 21.8 months	NCT02272855
**HF-10**	Nivolumab	Stage IIIB, IIIC, IVM1a melanoma	N/A	NCT03259425
**CAVATAK**	Ipilimumab	Uveal melanoma with liver metastasis	N/A	NCT03408587
**CAVATAK**	Pembrolizumab	Melanoma	N/A	NCT02565992
**ONCOS-102**	Pembrolizumab	Advanced or unresectable melanoma	N/A	NCT03003676

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
