# Peer review of "From Conventional Therapies to Immunotherapy: Melanoma Treatment in Review"

_cancers, 2020, doi:10.3390/cancers12103057_

Round 1
Reviewer 1 Report
The manuscript by Kuryk et al., "From Conventional Therapies to Immunotherapy: Melanoma Treatment in Review", describes current knowledge in the field of therapy of melanoma.
The authors discuss melanoma incidence and mortality worldwide, as well as its correlation with environmental risk factors and host-related risk factors.
The authors describe several conventional therapies for melanoma, including: excisional surgery, chemotherapy, as well as targeted therapies. Moreover, the authors describe current immunotherapy treatments based on immune checkpoint inhibitors, as well as oncolytic viruses.
Finally, the authors describe the most recent combinatorial treatments, including: oncolytic viruses with immune checkpoint inhibitors, as well as oncolytic viruses with chemotherapy and radiotherapy.
The manuscript is clearly written and all information provided is relevant and theory based, and well presented.
A few sentences must be revised (e.g.; “Immune checkpoint inhibitors are inhibitory signals…”) and a few typos need to be corrected (e.g.; “treatment regime…”).
Figure 1. The authors should correct errors and misspellings (i.e.; “advanced injectable Stage IIIb-IVM1c melanoma”, “involement”).
Author Response
Reviewer#1: The manuscript by Kuryk et al., "From Conventional Therapies to Immunotherapy: Melanoma Treatment in Review", describes current knowledge in the field of therapy of melanoma. The authors discuss melanoma incidence and mortality worldwide, as well as its correlation with environmental risk factors and host-related risk factors. The authors describe several conventional therapies for melanoma, including: excisional surgery, chemotherapy, as well as targeted therapies. Moreover, the authors describe current immunotherapy treatments based on immune checkpoint inhibitors, as well as oncolytic viruses. Finally, the authors describe the most recent combinatorial treatments, including: oncolytic viruses with immune checkpoint inhibitors, as well as oncolytic viruses with chemotherapy and radiotherapy. The manuscript is clearly written and all information provided is relevant and theory based, and well presented.
A: We thank the Reviewer for having considered our manuscript well constructed
Comment 1: A few sentences must be revised (e.g.; “Immune checkpoint inhibitors are inhibitory signals…”) and a few typos need to be corrected (e.g.; “treatment regime…”).
Answer to comment1: We thank the Reviewer for this comment. Sentences have been corrected throughout the manuscript.
Comment 2: Figure 1. The authors should correct errors and misspellings (i.e.; “advanced injectable Stage IIIb-IVM1c melanoma”, “involement”).
Answer to comment 2: We thank the Reviewer for this comment. Figure 1 has been revised.

Reviewer 2 Report
In the manuscript entitled “From Conventional Therapies to Immunotherapy: Melanoma Treatment in Review” Kuryk et al. describe the advantages of immunotherapy versus conventional therapies in melanoma. The review article clearly describes important aspects of combinatorial therapies involving checkpoint inhibitors and oncolytic viruses. However, the reported findings need to be discussed in more detail and supported by recent literature.
Major concerns:
- Authors should integrate information regarding the use of pembrolizumab and its use as first line therapy. Reference 48 (line 216) is obsolete.
- In the latest year atezolizumab has gained lots of attention due to its FDA approval for BRAF V600 unresectable or metastatic melanoma. The authors should discuss in more detail the importance of targeting PD-L1 in melanoma. Please revise figures accordingly.
Minor concerns:
- Please check the spelling of “checkpoint” along the manuscript.
- Figure 1 (line 192) should be cited later on in the manuscript.
- Figure 3 is not properly cited in the text (line 247).
- The authors should add a table summarizing the immuno-oncology agents and their combination therapies in melanoma.
Author Response
Reviewer#2: In the manuscript entitled “From Conventional Therapies to Immunotherapy: Melanoma Treatment in Review” Kuryk et al. describe the advantages of immunotherapy versus conventional therapies in melanoma. The review article clearly describes important aspects of combinatorial therapies involving checkpoint inhibitors and oncolytic viruses. However, the reported findings need to be discussed in more detail and supported by recent literature.
Major concerns:
Comment 1: Authors should integrate information regarding the use of pembrolizumab and its use as first line therapy. Reference 48 (line 216) is obsolete
Answer to commen1: We thank the Reviewer for this comment.
Information and references have been updated and can be found at page 6 lines 220-236.
The first monoclonal antibody against immune checkpoints to be discovered was ipilimumab, an anti-CTLA-4 antibody that has been firstly approved as a first-line treatment of metastatic melanoma (Wang S et al. Elife 2019). In the anti-PD-1 group there are other two common ICIs, pembrolizumab and nivolumab, both with indications for metastatic melanoma. Pembrolizumab has been the first anti-PD-1 monoclonal antibody that has been discovered, and with clinical trials KEYNOTE-001, KEYNOTE-002 and KEYNOTE-006 it has gained the first-line therapy indication for metastatic melanoma (Cowey CL et al J Immunother 2018). In particular, in trials KEYNOTE 006 AND KEYNOTE-002, which both presented comparative arms, patients treated with pembrolizumab significantly improved their progression-free survival (PFS), overall survival (OS) and overall response rates (ORR) relative to ipilimumab in ipilimumab-naive patients (KEYNOTE 006), and significantly improved PFS and ORR, but not OS (although OS data are immature), relative to chemotherapy in ipilimumab-refractory patients, who had also received BRAF/MEK inhibitor therapy if BRAF-mutation positive (KEYNOTE 002) (Deekes E.D Drugs 2016).
Pembrolizumab can to be administered as the first line therapy (BRAF wildtype melanoma) or after treatment with ipilimumab, in a combination with anti-CTLA-4 or in patients with BRAF mutations after treatment BRAF inhibitor such as vemurafenib, sorafenib and dabrafenib.
Comment 2: In the latest year atezolizumab has gained lots of attention due to its FDA approval for BRAF V600 unresectable or metastatic melanoma. The authors should discuss in more detail the importance of targeting PD-L1 in melanoma. Please revise figures accordingly
Answer to comment 2: We thank the Reviewer for this comment.
The PD-1/PD-L1 pathway plays an important immunosuppressive role, promoting self-tolerance and allowing tumor cells not to be recognized and then eliminated by immune system. Since many tumor cells express PD-L1, it has become of increasingly deeper interest to put more efforts in investigating this particular ligand and the effects of its targeting with monoclonal antibodies (Hayashi et al International Journal of Clinical Oncology 2020). As a result of these efforts, many immunotherapeutic agents targeting PD-1/PD-L1 have been approved by FDA, with indications for a wide array of malignancies (Gong J et al Journal for Immunotherapy of Cancer 2018).
Furthermore, on July 30, 2020, following analysis of IMspire150 phase III clinical trial (NCT02908672), the FDA approved atezolizumab (an anti-PD-L1 monoclonal antibody) in combination with cobimetinib and vemurafenib for unresectable, metastatic melanoma carrying BRAF V600 mutations (Gutzmer R et al. Lancet 2020). Even if pembrolizumab itself embodies a breakthrough in melanoma treatment, it is reasonable to think that atezolizumab could bring some new advantages if compared to the targeting of PD-1 exerted by pembrolizumab, such as the preservation of PD-L2 interactions with PD-1, which carries out the immune checkpoint functions that avoids autoimmune reactions during therapy, allowing for a more tolerable safety profile for this immunotherapeutic new drug (Seliger JCM 2019). We have added these aspects on pages 6 – 7 lines 235-241. The figures have been modified accordingly.
Atezolizumab in combination with cobimetinib and vemurafenib is also used for the patients with BRAF V600 mutation-positive unresectable or metastatic melanoma (first line therapy) (Figure 3) (IMspire150, NCT02908672) (Aris M et al Front Immunol 2015; Khair et alFront Immunol 2019; Larkin J et al, N Engl J Med 2015;Zimmer Let al, Eur J Cancer 2017). Therefore, it is reasonable to suppose that atezolizumab could bring some new advantages if compared to the targeting of PD-1 exerted by pembrolizumab, such as the preservation of PD-L2 interactions with PD-1 which carries out the immune checkpoint functions that avoids autoimmune reactions during therapy, allowing for a more tolerable safety profile for this immunotherapeutic new drug (Seliger B, J Clin Med 2019).
Minor concerns:
- Please check the spelling of “checkpoint” along the manuscript.
A: We have corrected the mistakes along the manuscript
- Figure 1 (line 192) should be cited later on in the manuscript.
A: Figure 1 has been now included in the revised version of this manuscript as Figure 2 and has been cited along the manuscript
- Figure 3 is not properly cited in the text (line 247).
A: Figure 3 has been now included as Figure 1 and cited on page 5 line 194
- The authors should add a table summarizing the immuno-oncology agents and their combination therapies in melanoma
A: We thank the Reviewer for this comment. A table summarizing the immune-oncology agents and their combination for melanoma treatment has been included as Table 1 on page 3

Reviewer 3 Report
overall/ the introduction is nicely written and well explained. from page 5 on there are issues that should be resolved which I've tried to address there:
Pg 5 line 215: sorafenib is not anti-BRAF blocker used for melanoma; also pembrolizumab/nivolumab can be given before ipilimumab (which is mostly done in clinical practice)
Pg 6 line 230: notable: explain percentage like you do in next paragraph
Pg 6 line 233: add trial names as you do for previous paragraph
Pg 8 line 327: there is much more preclinical research: I would propose to either talk about all preclinical research papers or not to explain any preclinical papers. As the first author is employee of a company it seems biased (although it might not be)
Pg 8 line 337: there are more OVs in clinical development: why do you only talk about this one? Please make an overview of all unless you define a reason why that one would be the only interesting one
Pg 8 line 338: I dont understand this phrase (deep …. Potential)
Pg 8 line 344 add reference please
Pg 8 line 347 and further: I dont understand: according to you do OV create neoantigen or just release neoantigen in the TME so that they can be loaded on actived DCs? Please clarify
Pg 9 line 355 this is why…: maybe add a paragraph what questions have to be answered by clinical trials and how this should be addressed
Pg 9 line 357 and further: please write with more clarity, can also be shortened. Also for pembrolizumab there is not always PDL1 staining required as is the case for melanoma… Also which biomarker are you thinking of for chemotherapy? Acc to my knowledge this is not done in clinical practice.
Page 9 line 374: chemo is not standard in melanoma: a bit confusing: is the goal of the review to write about melanoma therapy or about OV therapy in general?
In vitro/in vivo should be written in italic
Page 10 line 421: preclinical cell line results… for me not enough evidence to call this consistent conclusions. In my opinion there is a lot of discussion around the adjuvancy effect of radiotherapy. If no clinical results, please state so and maybe mention if there are mouse models.
Author Response
Point by point replies
Reviewer#3: overall/ the introduction is nicely written and well explained. from page 5 on there are issues that should be resolved which I've tried to address there:
Comment 1: Pg 5 line 215: sorafenib is not anti-BRAF blocker used for melanoma; also pembrolizumab/nivolumab can be given before ipilimumab (which is mostly done in clinical practice).
Answer to the comment 1: We thank the Reviewer for the comment. Indeed, the Sorafenib is a kinase inhibitor, members of the MAPK pathway and receptor tyrosine kinases, including VEGF-R2. Sorafenib, carboplatin and paclitaxel (SCP) has anti-tumor activity in melanoma patients, but no association was found between response and activating B-RafV600E mutations. We have corrected the text by removing the Sorafenib. Additionally, we do agree that the combination of the pembrolizumab/nivolumab with ipilimumab showed effective anti-cancer efficacy in melanoma patients (however can be toxic for melanoma patients). Advanced melanoma patients who received nivolumab plus ipilimumab or anti-PD1 alone showed sustained long term 5 years overall survival compared to the percentage of patents who received only ipilimumab NCT01844505 (Larkin J et al, Lancet 2019). It has been also shown that the ipilimumab after failure to anti-PD-1 therapy induces comparable response rates as treatment of the naïve patients. Therefore, ipilimumab can be considered an option for those cancer patients who progressed on prior anti-PD-1 treatment. Overall response (OR) rate of ipilimumab plus nivolumab after failure to anti-PD-1 is lower compared to treatment-naïve patients (Zimmer L et al, European Journal of Cancer 2017). Clinical studies also have shown anti-tumor activity with a favorable toxicity profile in patients treated with low-dose ipilimumab (1 mg/kg) and standard-dose pembrolizumab (2 mg/kg) (Kirchberger MC et al Oncotarget 2018).
The updated text ‘’ Pembrolizumab can to be administered after treatment with ipilimumab, in a combination with anti-CTLA-4 or in patients with BRAF mutations after treatment with a BRAF inhibitor such as vemurafenib and dabrafenib’’ is available on the pages 6-7 (lines 232-241).
Comment 2: Pg 6 line 230: notable: explain percentage like you do in next paragraph
Answer to the comment 2: We have added request information about the clinical efficacy data. The corrected text can be found on the page 7 (lines 272-273).
The tumor responses according to the Response Criteria in Solid Tumors (RECIST) criteria varied from 5.7% to 11.0% in the anti-CTLA-4 treatment arms. The median overall survival (OS) was improved to 10 months for the anti-CTLA-4 monotherapy arm as compared to 6.4 months for the peptide vaccine-alone arm (HR 0.68; p < 0.001 (Rogiers A et al, Journal of Oncology 2019), CA184-002, NCT00094653). The five-year survival rate was 18.2% (95% CI, 13.6% to 23.4%) for patients treated with anti-CTLA-4 + dacarbazine vs. 8.8% (95% CI, 5.7% to 12.8%) for patients treated with placebo plus dacarbazine (p = 0.002, CA184-024, NCT00324155) (Maio M et al Journal of Clinical Oncology 2015).
Comment 3: Pg 6 line 233: add trial names as you do for previous paragraph
Answer to the comment 3: The text was reformatted, and trial number added. The corrected text can be found on the page 8 line 274.
Comment 4: Pg 8 line 327: there is much more preclinical research: I would propose to either talk about all preclinical research papers or not to explain any preclinical papers. As the first author is employee of a company it seems biased (although it might not be).
Answer to the comment 4: We thank the Reviewer for the comment. We have updated the section and provided more examples of the pre-clinical and clinical findings on the combinatory therapy with oncolytic vectors and CPIs in melanoma therapy. The corrected text can be found on the pages 9-10 (lines 364-417).
The Hemminki group exploited a murine model of melanoma to establish the mechanism under the combination of the anti-PD-1 antibody with the oncolytic viruses encoding for TNFα and IL-2 (Cerveira et al Oncoimmunology 2018). What emerged from the combination therapy was a marked increase in intratumoral CD8+ T cells and a statistically significant tumor growth suppression, along with increased survival in animals. Researchers reported complete tumor regression after the course of the combinatory therapy. This preclinical research provides the rationale for a clinical trial where oncolytic adenovirus coding for TNFa and IL-2 (TILT-123) is used in melanoma patients receiving an anti-PD-1 antibody NCT04217473) (Cerveira et al Oncoimmunology 2018).
Thomas et al. reported development of a new fusion-enhanced oncolytic immunotherapy platform based on herpes simplex virus type 1. Researchers developed various oncolytic vectors expressing e.g. GMCSF, an anti-CTLA-4 antibody-like molecule. Anti-cancer assessment was performed in vivo and in nude mouse xenograft models (melanoma, lymphoma, gliosarcoma). The combination therapy with the virus expressing GALV-GP-R- and mGM-CSF and an anti-murine PD1 antibody showed improved anti-tumor effects compared to the control. The treatment of mice with derivatives of this virus coding for anti-mCTLA-4, mCD40L, m4-1BBL, or mOX40L showed enhanced anti-cancer efficacy in un-injected tumors (abscopal effect) (Thomas S et al, Journal for Immunotherapy of Cancer 2019).
Also, in our previous study we have investigated the canti-cancer potency of ONCOS-102 and pembrolizumab in the humanized melanoma mouse model. Humanized mice engrafted with A2058 melanoma cells showed significant tumor volume reduction after ONCOS-102 treatment. The combination of anti-PD1 with the virus further reduced tumor volume, while pembrolizumab alone did not show therapeutic benefit by itself (Kuryk et al, Oncoimmunology 2019). Systemic abscopal was also observed when combining oncolytic adenovirus and checkpoint inhibitor in a humanized NOG mouse model of melanoma (Kuryk L et al, JMV 2019). These data support the scientific rationale for the ongoing clinical study of combination therapy of ONCOS-102 and pembrolizumab for the treatment of melanoma (NCT03003676).
Currently, there are many oncolytic vectors are under development and investigation in melanoma: coxsackieviruses, HF-10, adenoviruses, reoviruses, echoviruses, and Newcastle disease viruses. Therefore, it is probable that oncolytic vectors will have long-term application in the treatment of advanced melanoma not only as a monotherapy but as a part of combinatory therapies (Jahjefendic et al, Biomedicines 2020). T-VEC is the first oncolytic vector approved for the melanoma treatment. Reported data have shown improved therapeutic responses to T-VEC in combination with immune checkpoint blockade in patients with melanoma without additive toxicity (LaRocca CA et al. American Journal of Clinical Dermatology 2020). T-VEC combined with anti-PD-1 based immunotherapy for unresectable stage III-IV melanoma showed an overall response rate for on-target lesions of 90%, with 6 patients resulting in a complete response in injected lesions (NCT02263508) (Sun L et al. Journal of Immunotherapy of Cancer 2018). Also, the treatment with T-VEC in patients with advanced melanoma with disease progression following multiple previous systemic therapies (vemurafenib, metformin, ipilimumab, dabrafenib, trametinib, and pembrolizumab) showed signs of anti-cancer effect, and provides potential clinical and immunotherapeutic utility of T-VEC application (Jason C et al, Melanoma Research 2018).
CAVATAK, an oncolytic immunotherapy, is an oncolytic strain of Coxsackievirus A21 (CVA21). The virus infects ICAM-1 expressing tumor cells, resulting in cell lysis, and anti-tumor immune response. The Phase II CALM study investigated the efficacy and safety of CVA21 in patients with advanced melanoma (NCT01227551). The treatment with CAVATAK resulted in elevation of the immune CD8+ T cell infiltrates within the tumor (5 of 6 patients), and increased expression of PD-L1+ cells. It was also reported that the virus was able to reconstitute immune cell infiltrates in lesions resistant to immune-checkpoint blockade (Andtbacka et al Journal of Immunotherapy of Cancer 2015). The combinatory therapy trials have been conducted where CAVATAK was administered with ipilimumab (NCT02307149) or pembrolizumab (NCT02565992). The treatment with CAVATAK and anti-CTLA-4 has shown durable response with minimal toxicity. The preliminary ORR rate for the ITT population of 50.0% is higher than published rates for either agent used alone (CAVATAK: ~28% and ipilimumab: ~15-20%) in advanced melanoma patients (Curti B et al, Cancer Research 2017). Among the evaluable patients (intratumoral CAVATAK and systemic pembrolizumab in advanced melanoma patients), the ORR was 73% (8/11). The DCR (CR+PR+SD) was 91% (10/11). In patients with stage IVM1c disease, the ORR and the DCR is 100% (5/5). Combination therapy of the virus1 and anti-PD1 may present a new startegy for the treatment of patients with injectable advanced melanoma (CAPRA clinical trial) (Schmid P et al, Cancer Research 2017)
Another oncolytic adenovirus that has been investigated in combination with pembrolizumab is ONCOS-102 (AdV5/3-Δ24-GM-CSF), which is now under clinical trial (NCT03003676) to investigate its safety and efficacy, supported by preclinical data showing increased CD8+ T cell infiltration in tumor mass upon viral administration [80]. The therapeutics efficacy and safety of the virus was previously tested in C1 study (NCT01598129). The treatment with the virus was safe and well tolerated at the tested doses. Therapy resulted in infiltration of CD8+ T cells to tumors and up-regulation of PD-L1, highlighting the potential of ONCOS-102 as an immunosensitizing agent for combinatory therapies with checkpoint inhibitors (Ranki T et al, Journal of Immunotherapy of Cancer 2016). Therefore, providing a scientific rationale for the combinatory therapy with CPIs.
Comment 5: Pg 8 line 337: there are more OVs in clinical development: why do you only talk about this one? Please make an overview of all unless you define a reason why that one would be the only interesting one
Answer to the comment 5: As mentioned in the answer to the question 4 above. We have updated the manuscript with a broaden description of preclinical findings and clinical studies, represented by various oncolytic vectors.
Comment 6: Pg 8 line 338: I dont understand this phrase (deep …. Potential)
Answer to the comment 6: We have corrected the confusing sentence. The new text has been added and is available on the page 11 lines 449-451.
To date, approximately one third of all clinical trials concerning OVs have investigated a combinatorial approach with at least one ICI (Russell L et al BioDrugs 2019). Therefore, it is expected that oncolytic viruses have the capability to promote a ‘hotter’ immune microenvironment which can improve the efficacy of ICI (Chiu M et al Expert Opinion on Biological Therapy 2020).
Comment 7: Pg 8 line 344 add reference please
Answer to the comment 7: We thank the Reviewer for this comment. Reference has been included
Comment 8: Pg 8 line 347 and further: I dont understand: according to you do OV create neoantigen or just release neoantigen in the TME so that they can be loaded on actived DCs? Please clarify
Answer to the comment 8: We have removed the confusing sentence (page 11, line 435).
Comment 9: Pg 9 line 355 this is why…: maybe add a paragraph what questions have to be answered by clinical trials and how this should be addressed
Answer to the comment 9: ICIs have contributed to revolutionize cancer treatment. Nevertheless, the best response rates to these agents do not exceed 35% to 40% (Zamarin D et al. Sci. Transl. Med. 2014; Callahan et al. Immunity. 2016;44:1069–1078). Therefore, the goal of combining OVs with ICIs is to enhance clinical efficacy. Oncolytic vectors are used in order attract the immune cells into the lesion, prime anti-tumor immune responses by development of innate and adoptive anticancer immunity. In turn, CPI therapy will prevent inhibition of activates cancer specific T cells. It is expected that those two agents can result in synergistic or additive anti-cancer effect. Interestingly, research group has demonstrated that local oncolytic virus injection can modulate tumor‐specific CD8+ T‐cell responses rendering distant tumors susceptible to immune checkpoint inhibitor therapy (Woller N et al., Mol Ther 2015). Therefore, due to the preclinical success of this combination therapy, there is huge interests in clinical trials: results obtained from patients who have progressed after immune checkpoint inhibition (e.g. NCT 03003676) could shed the light on OV’s role in overcoming resistance to immunotherapy. By elucidating the potential of the combination of OVs and checkpoint inhibitors, further development in treatment regimens employing these novel therapeutic agents could be beneficial for patients. These aspects have been reported on page 11 lines 443-445.
Comment 10: Pg 9 line 357 and further: please write with more clarity, can also be shortened. Also for pembrolizumab there is not always PDL1 staining required as is the case for melanoma… Also which biomarker are you thinking of for chemotherapy? Acc to my knowledge this is not done in clinical practice.
Answer to the comment 10: Apart from combinatorial strategies, another aspect concerning the use of ICIs is often investigated to reach some improvement—the response predictions with biomarkers. There are several biomarkers associated with the response of ICIs, some of which have been approved and are currently being exploited to predict the response rate in patients before treatment begins, while others are under further study to establish whether they have a strong correlation with the extent of patients’ responses to ICIs. The most important predictive biomarker for anti-PD-1/PD-L1 antibodies is PD-L1 expression (Cottrell The Cancer Journal 2018;Ming Yi et al Molecular Cancer 2018; Davis A et al. Journal for immunotherapy of Cancer 2019; Shukuya et al Journal of Thoracic Oncology 2016), which is evaluated by immunohistochemistry. PD-L1 expression by cancer cells is recognized as both a prognostic and predictive biomarker in patients with cutaneous melanoma. Approx. 35% of cutaneous melanomas express PD-L1(Kaunitz GJ et al, Laboratory Investigation 2017). The PD-L1 immunohistochemistry (IHC) has been approved by FDA as a complement diagnostic to select patients with non-small-cell lung carcinoma (NSCLC) suitable for pembrolizumab therapy. Nevertheless, absence of PD-L1 does not necessarily translates into a poor response to anti-PD-1/PD-L1 inhibitors. Some patients with low PD-L1 expression exhibits clinical efficacy. However, further efforts are still needed to improve the clinical use of PD-L1 expression as biomarkers (Yan X et al Front Pharmacol 2018).
Comment 11: Page 9 line 374: chemo is not standard in melanoma: a bit confusing: is the goal of the review to write about melanoma therapy or about OV therapy in general?
Answer to the comment 11: We agree with the Reviewer that chemotherapy is not a standard therapy for the treatment of melanoma. The aim of the paragraph was to provide a prospect on the potential future therapies, utilizing the oncolytic virotherapy with chemotherapy, that might be tested clinically based on the available preclinical findings (in vitro and in vivo). Chemotherapeutic agents used in combination with oncolytic viruses can potentiate their cytotoxic mechanisms. Counteracting immunological barriers can improve the persistence of viruses and/or weaken the immunosuppressive forces within the tumor microenvironment (Nguyen et al Front Oncol 2014). We have updated the header name - OVs with Chemotherapeutic Agents – future prospects.
In vitro/in vivo should be written in italic
Answer: Italic has been used for in vitro/in vivo
Comment 12: Page 10 line 421: preclinical cell line results… for me not enough evidence to call this consistent conclusions. In my opinion there is a lot of discussion around the adjuvancy effect of radiotherapy. If no clinical results, please state so and maybe mention if there are mouse models.
Answer to the comment 12: The aim of the paragraph was to provide a prospect on the potential future therapies, utilizing the oncolytic virotherapy with radiotherapy, that might be tested clinically based on the available preclinical findings (in vitro and in vivo). The paragraph was reformatted and is available on the page 13, lines 540-547.
Twigger et al. tried to combine an oncolytic reovirus with radiation therapy in a variety of melanoma cell lines, observing that the combination yielded a statistically significant enhancement of viral cytotoxicity without affecting reoviral replication rates, but with an increase in apoptosis of cancer cells (Twigger K et al. Clinical Cancer Research 2008). In another preclinical study, Kyula et al. investigated the combination of an oncolytic Vaccinia virus and radiotherapy in BRAF-mutated, Ras-mutated and wild type melanoma cell lines. Results showed that in melanoma cells that carried V600D or V600E BRAF mutations there had been an increased apoptosis (Kyula JN et al Oncogene 2013). Also, the combination of reovirus and radiation has shown to increase the tumor growth delay of the melanoma xenografts in the treated animals, and significantly improve the overall survival rate compared to the treatment with either of the individual therapies (Zhang T et al The Open Virology Journal 2017). Importantly, Ras mutation is one of the driver mutations for melanoma and is associated with radio-resistance (Gupta AK et al, Int. J. Radiation Oncology Biol. Phys. 2003). However, some viruses like: reovirus, VSV and HSV have been able to selectively target the Ras mutated melanoma cells and mediate cell death (Noser J Mo Ther 2007). Therefore, oncolytic vectors able to lyse the radiation-resistant melanoma cells can exhibit a complementary therapeutic effect to radiotherapy. There are many ongoing attempts to find the optimal way to combine these two strategies to maximize the antitumor effect preclinically. More investigations are needed to understand how to exploit this combination in the complex context of metastatic unresectable melanomas and their application in clinics.

Round 2
Reviewer 2 Report
In the revised version of the manuscript, Kuryk et al. replied to all reviewer concerns.
Considering the adjustments made by the authors in the revised version, it would be preferable to modify the abstract accordingly. In particular the authors should not mention one specific oncolytic virus therapy but they should rather generalize the concept by citing the different oncolytic virus and immunotherapy approaches.
Author Response
Point by point replies
Reviewer#2:
Comment: In the revised version of the manuscript, Kuryk et al. replied to all reviewer concerns. Considering the adjustments made by the authors in the revised version, it would be preferable to modify the abstract accordingly. In particular the authors should not mention one specific oncolytic virus therapy but they should rather generalize the concept by citing the different oncolytic virus and immunotherapy approaches.
Answer: We thank the Reviewer for nice comment. We have modified the abstract accordingly. Please find the new sentences on page 1 lines 25-27 and line 30.
Reviewer 3 Report
all queries were nicely addressed by the authors. the ms improved and can be accepted. maybe just when proofreading by native englisch; congrats with the quick improvements
Author Response
Comment:All queries were nicely addressed by the authors. the ms improved and can be accepted.maybe just when proofreading by native english; congrats with the quick improvements
Answer: We are pleased with the Reviewer comment and statement that all queries were nicely addressed by the authors. We also want to inform that the manuscript has undergone English language editing by MDPI. The text has been checked for correct use of grammar and common technical terms, and edited to a level suitable for reporting research in a scholarly journal. Please find enclosed the certificate.
